# Accessibility of Public Sector Institutions for People with Special Needs in Polish Regions

Marcin Janusz [1], Marek Piotrowski [1,*], Emilia Kwiatkowska [2], Mariola Grzybowska-Brzezińska [3] and Kamil Maciuk [4]

1    Department of Economic Policy, Faculty of Economic Sciences, University of Warmia and Mazury in Olsztyn, Oczapowskiego 4, 10-719 Olsztyn, Poland; marcin.janusz@uwm.edu.pl
2    Spatial Planning Office of the Lodz Region in Lodz, Regional Territorial Observatory, al. Piłsudskiego 12, 90-051 Lodz, Poland; emodranka@wp.pl
3    Department of Market and Consumption, Faculty of Economic Sciences, University of Warmia and Mazury in Olsztyn, Cieszyński Sq. 1/327, 10-720 Olsztyn, Poland; margrzyb@uwm.edu.pl
4    Department of Integrated Geodesy and Cartography, AGH University, 30-059 Krakow, Poland; maciuk@agh.edu.pl
*    Correspondence: mpiotrowski@uwm.edu.pl

**Abstract:** Social inclusion is one of the important conditions for sustainable socio-economic development. However, one of the paths leading to social inclusion is to ensure a high level of accessibility of institutions for people with special needs. The study described in the present paper examined the accessibility of Polish public sector bodies based on data from government reports (comprehensive study). Accessibility is a feature that should be offered as a complementary service offered to both individuals and legal entities during epidemic emergencies and beyond. The limited accessibility of public institutions is a significant obstacle to the full well-being of the population. Indicating differences in the spatial dispersion of this phenomenon in Poland was the basis for undertaking research for this article. Three areas of accessibility were tested: physical (architectural) accessibility, digital (web) accessibility, and ICT accessibility. A synthetic measure of accessibility was constructed for the study and used to rank Polish voivodeships (provinces, NUTS 2) from highest to lowest. Clustering was used to identify similar regions. The highest- and lowest-scoring regions were the Mazowieckie voivodeship (capital city—Warsaw) and the Podkarpackie voivodeship (capital city—Rzeszów), respectively. Legal accessibility requirements are the biggest driver of further accessibility improvements for voivodeships.

**Keywords:** accessibility; public institutions; digital exclusion; digital divide; information society; TOPSIS; NUTS 2

## 1. Introduction

Socioeconomic development tends to increase expectations among the public as to the quality and standard of products and services they procure. Social inclusion has also been cited as a major factor [1–3], noting that all social groups should have equal access to such products and services. This begets initiatives aimed at improving service availability and quality, which should extend to public institutions [4,5].

The COVID-19 pandemic has profoundly changed the way societies and economies function. Universal lockdowns [6], coupled with remote learning [7] and work-from-home arrangements [8], have transformed professional activity and social relations. Two knock-on effects are particularly noteworthy. First, the transition to digital space has highlighted the problem of digital exclusion [9,10]. Limitations in access to ICT infrastructure and inability (or unwillingness) to use digital devices and systems suddenly became a major obstacle to satisfying one's needs. Secondly, the pandemic forced many institutions (especially public ones) to revise the existing conditions and criteria for accessibility. This relates primarily to

digital and ICT accessibility, though accessibility in the physical (architectural) sense has been affected as well.

The literature on the subject describes modern societies as information societies [11]. The term started to emerge once it became apparent that technological advancement and information flow (among other less significant factors) were transforming many areas of social life. These developments have removed barriers of distance and allowed information to be used in all areas of social and economic life. Grodzka [12] notes that Poland was quite late to start building an information society along European guidelines and did so mainly out of a desire to join the European Union.

One of the major challenges of digital transformation has been combating digital exclusion [13–17]. The latter term, according to van Dijk [18], should extend beyond a simple delineation between those with and without Internet access [19]. Thus, he distinguishes four levels of access to new media, which are a confluence of barriers and expectations faced by an individual. These include physical access (computer ownership and Internet access), motivation to use new technologies and skills (strategic, informational, and operational), and the different ways of using new technologies [20]. Batorski [21] draws on this to put forward a two-pronged approach to eliminating digital exclusion—pursuing infrastructure-related activities and developing soft competencies within society [22]. After all, whereas digital exclusion may have numerous [23] socioeconomic determinants (age [24], gender [25,26], education [27,28], material status [29], and place of residence [30,31]), it is individual preferences and attitudes that ultimately drive success.

Given that ICT solutions have become embedded in everyday life, it was necessary to develop a theoretical framework to define and operationalize this problem. Accessibility was the first concept to be defined. Nowadays, it is considered to be a feature of an environment that allows persons with functional and cognitive disabilities equitable use of this environment. For many people, such accessibility is crucial for social and economic engagement [32].

This qualitative definition of accessibility is still often ignored or marginalized on the misguided grounds that so-defined accessibility only benefits a small minority. This view may stem from inadequate awareness of real needs in this regard, which are driven in part by ageing populations with an increasing number of people with disabilities/special needs due to age, ability, illness, etc. For an increasing number of people, accessibility has become a way to guarantee an independent, unassisted, and better life. Its significance is growing each year and cannot be overlooked, particularly in Poland, as a CEE country with the experience of an economy restricted from open market solutions.

To meet the challenges and expectations of the information society, quality and accessibility standards had to be developed for services. Unified accessibility criteria had to be created to facilitate digital technology use and the flow of information. These were ultimately formalized in the form of WCAG (Web Content Accessibility Guidelines)—a set of criteria delineating how to design websites and applications friendly to the visually and hearing impaired, physically disabled, and mentally/cognitively disabled. WCAG-compliant websites and mobile apps are considered digitally accessible [33].

These guidelines should be followed by public sector instructions. Entities are currently obliged to follow the requirements set in applicable legislation and report on the findings of such analyses, e.g., for international comparisons. Studies can approach accessibility from the point of view of the implementers, as has been performed both in developing [34] and developed economies [35,36]. Other papers have tested accessibility for selected social groups. For example, Botelho [37] investigated web accessibility for the disabled, whereas Roszewska [38] considered GIS as a tool for improving everyday life. Rodaković et al. focused primarily on the physical accessibility of public institutions for persons with a locomotor disability [39], whereas Tsatsou [40] included ethnic minorities and the elderly in their analyses. Another avenue of research has focused on public employees (especially social workers) to identify and overcome barriers to web accessibility [22],

including those created by the pandemic [41–43]. Finally, some researchers have set out to verify the web accessibility of public sector bodies [5,44] and the quality of information published on their websites [45].

The present study sought to assess the accessibility of Polish public bodies on the regional level. For a long time, this was hindered by a dearth of comprehensive sources. However, the Ministry of Development Funds and Regional Policy has since published a dedicated report that provides the missing data. Given the wide gaps in socioeconomic development between Polish regions [46], it seemed reasonable that there would also be disparities in accessibility to public sector bodies. The awareness of the close relationship between social inclusion and sustainable socio-economic development even necessitates the need to deepen knowledge about the level of accessibility of various institutions for people with special needs.

The present study consists of four main sections. The introduction describes the research problem and reviews the literature on the subject. The second part presents the sources of data used in the analysis as well as the analytic methods used to develop an original synthetic measure of public institution accessibility in Poland. The third part presents the results and compares them against similar research. The final part provides a summary and conclusions.

## 2. Materials and Methods

The present study draws on data from "Status report on providing accessibility to people with special needs" ("Raport o stanie zapewniania dostępności osobom ze szczególnymi potrzebami przez podmioty publiczne w Polsce") prepared by the Ministry of Development Funds and Regional Policy [32] and facilitated by the United Nations Convention on the Rights of Persons with Disabilities. Under this framework, Poland has been implementing the Accessibility Plus 2018–2025 programme. Furthermore, two acts adopted in 2019 (the Act of 4 April 2019 on Digital Accessibility of Websites and Mobile Applications of Public Entities [47] and the Act of 19 July 2019 on Providing Accessibility to People with Special Needs [48]) impose specific accessibility requirements (mainly) on public sector bodies. As explained in the report [32]: "The acts regulate the digital, physical (architectural) and ICT aspects of public-sector body functioning so as to facilitate unassisted access by persons with disabilities or those temporarily in need of aid".

The problem of public sector body accessibility has not been researched so far, mainly due to objective difficulties in assessing accessibility standards and requirements, among other challenges. However, WCAG-based regulations have since been implemented, and public bodies have been obliged to report accessibility status, providing a legal and formal basis for accessibility studies. Due to the aforementioned difficulties, the present paper is the first such comprehensive study in Poland.

The data provided in the report relate to public body accessibility as of 1 January 2021 and have been collected through online surveys from over 57,500 Polish entities (Figure 1). Over 67,000 websites/apps and 114,000 buildings have been tested for accessibility in over 3700 towns/villages for the needs of this survey.

For the purposes of the present paper, accessibility is defined as a feature of an environment that enables persons with functional and cognitive difficulties to benefit from the environment on an equitable basis, which is crucial for social and economic engagement. Public body accessibility is tested in three respects: the physical (architectural), the digital, and the technological (ICT). Each has its own standards and requires components that are presented in Tables 1–3.

The report thus provides a respectable collection of accessibility data for Polish public sector entities. For the purposes of this paper, the analysis focused on the accessibility data that were reported on a by-voivodeship basis (NUTS level 2). The aim was to construct a synthetic accessibility measure that could show regional differences in accessibility between voivodeships. In other words, the study sought to assess public entity accessibility on the regional level.

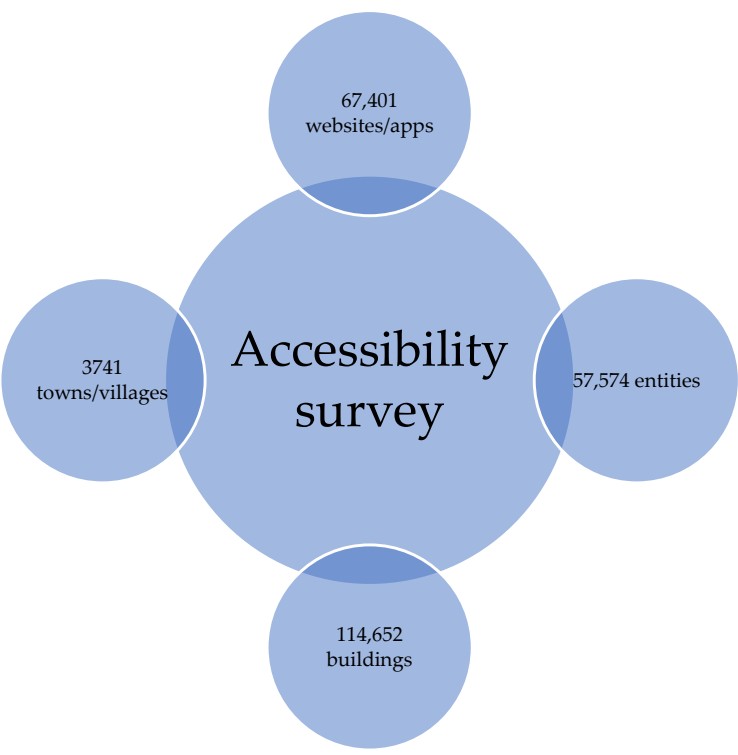

**Figure 1.** Characteristics of the survey.

**Table 1.** Physical standards of accessibility.

| Physical Standards of Accessibility (Expected) | |
|---|---|
| 1 | barrier-free horizontal and vertical navigation indoors |
| 2 | use of fixtures, fittings, aids, equipment and architectural measures that enable free access to all rooms in the building (with the exception of service rooms) |
| 3 | information on the room layout in the building—visual and tactile or audio- (voice-) based at minimum |
| 4 | allowing entry to persons with guide dogs |
| 5 | ensuring evacuation procedures or other means of rescue for persons with special needs |

Source: own elaboration based on [47,48].

**Table 2.** Digital standards of accessibility.

| Digital Standards of Accessibility (Expected) | | |
|---|---|---|
| 1 | a Public Information Bulletin website for the entity | |
| 2 | incorporation of the following components and features on the website and/or mobile app: | |
| | (a) | contact details of the public entity and a link to the Public Information Bulletin website of the entity (if it is obliged to maintain one under separate regulations) |
| | (b) | tools for contacting the public entity |
| | (c) | navigation |
| | (d) | an accessibility statement for the website and/or mobile app of the public entity, hereinafter referred to as the "accessibility statement" |
| | (e) | emergency information within the meaning of Article 3 (1) of the Act of 26 April 2007 on Crisis Management (Journal of Laws of 2022, items 261, 583 and 2185) and other public safety information promulgated by the public entity |
| | (f) | official documents, template contracts and/or other model documents required to enter into civil law obligations |

Source: own elaboration based on [47,48].

**Table 3.** ICT standards of accessibility.

| ICT Standards of Accessibility (Expected) | |
| --- | --- |
| 1 | service that includes communication aids (sign language and other means of communication) and/or remote (online) access to a translation service via websites and apps, i.e., the entity must support contact via phone, electronic mail, SMS/MSS text messages, online instant messaging software, etc., through audiovisual communication, including by means of online instant messaging software, fax communication, sign language translation–face-to-face, online, or a sign-language interpreter |
| 2 | use of equipment or other technological measures to accommodate the hard of hearing (induction loops, FM systems, IR systems, and/or other equipment/systems) |
| 3 | a "what we do" notice on the website in machine-readable text, in Polish Sign Language (video recording) and in easily readable text |
| 4 | if a person with special needs so requests—providing communication via the requested means |

Source: own elaboration based on [47,48].

The synthetic accessibility measure was calculated using TOPSIS [49], commonly used in economics research [50–52]. The method weighs decision alternatives by measuring their distances from two reference points—the PIS (Positive Ideal Solution) and the NIS (Negative Ideal Solution). In this system, the best decision is the one closest to the PIS and furthest from the NIS. The procedure is performed in several steps. The first step is to build a decision matrix $X = [x_{ij}]$ and calculate the weight vector $w = [w_1, \ldots, w_n]$, where

$$w_1 + \ldots + w_n = 1. \tag{1}$$

The next step is to build a normalized decision matrix $N = [z_{ij}]_{mxn}$, where $z_{ij}$ is the value of the normalized decision alternative assessment, according to the formula:

$$z_{ij} = \frac{x_{ij}}{\sqrt{\sum_{i=1}^{m}\left(x_{ij}^2\right)}} \tag{2}$$

where $i = 1, \ldots, m$, and $j = 1, \ldots, n$.

The next step is to build a normalized weighted decision matrix:

$$W = [v_{ij}]_{mxn}, \text{ where } v_{ij} = w_j z_{ij}. \tag{3}$$

The positive ideal solution $A^+$ and negative ideal solution $A^-$ are then determined in the form of:

$$A^+ = \left[v_1^+, \ldots, v_n^+\right] \text{ where } v_j^+ = \begin{cases} \max v_{ij}, & v_{ij} \in Z \\ min\ v_{ij}, & v_{ij} \in S \end{cases} \tag{4}$$

$$A^- = \left[v_1^-, \ldots, v_n^-\right] \text{ where } v_1^- = \begin{cases} \min v_{ij}, & v_{ij} \in Z \\ max\ v_{ij}, & v_{ij} \in S \end{cases} \tag{5}$$

The next step is to calculate the distance $(d_i^+)$ of the i-th decision alternative from $A^+$ and the distance $(d_i^-)$ of the i-th decision alternative from $A^-$,

$$d_i^+ = \sqrt{\left(v_{ij} - v_j^+\right)^2}, \ d_i^- = \sqrt{\left(v_{ij} - v_j^-\right)^2} \tag{6}$$

where $i = 1, \ldots, m$.

This allows the value of the synthetic assessment measure (the global score) to be calculated for the *i*-th decision alternative, according to the formula:

$$T_i = \frac{d_i^-}{d_i^+ + d_i^-} \tag{7}$$

where $i = 1, \ldots, m$. Value of $T_i \in [0, 1]$.

Finally, the decision alternatives were ranked in a decreasing order of the synthetic assessment score. The higher the score, the higher the alternative was ranked.

For data readability, the assorted objects (in this case—the voivodeships) were then assigned a specific class, resulting in a four-class classification of voivodeships according to their synthetic accessibility score. The groups were classified as per the formula:

Group 1: (max; max-k); Group 2: <max-k; max-2k); Group 3: <max-2k; max-3k); Group 4: <max-3k, min)

where

max—the highest value of the synthetic measure;
min—the lowest value of the synthetic measure;
k = range/number of groups.

This evaluation is supplemented by an analysis of similarities in accessibility scores between the voivodeships. Grouping was carried out using classification methods in order to identify those clusters of objects that were the most similar (homogeneous) in their structure of observed values—in this case, the synthetic measures of accessibility. The clusters should show strong differentiation between the groups but maximum homogenization within the groups ([53], p. 66).

Ultimately, Ward's method was used for agglomerate hierarchical clustering, where the starting number of clusters is equal to the number of test objects. The criterion for classifying units into higher-order clusters (groupings) was the minimal variance of values for parameters ([54], p. 122) that were used as segmentation criteria relative to the value of the clusters generated in the successive steps. In effect, the objects included in a given group are as similar as possible in terms of the analyzed parameters. Subsequent iterations are determined by the distance ($d_{ip}$) between the newly generated cluster and the remaining clusters, as derived from the following formula ([55], p. 278):

$$d_{ip} = \frac{n_i + n_k}{n_i + n_j + n_k} d_{ik} + \frac{n_j + n_k}{n_i + n_j + n_k} d_{jk} - \frac{n_k}{n_i + n_j + n_k} d_{ij}$$

where

$n_i$—size of cluster *I*;
$n_j$—size of cluster *j*;
$n_k$—size of cluster *k*;
$d_{ik}$—distance between original cluster *i* and cluster *k*;
$d_{jk}$—distance between original cluster *j* and cluster *k*;
$d_{ij}$—distance between original cluster *i* and original cluster *j*.

Ward's method is widely accepted due to its theoretical value and good simulation results, offering excellent grouping performance with highly homogenous clusters. Another advantage is the high readability of results, which are presented in a dendrogram form.

It is impossible to formulate a fully comprehensive set of accessibility metrics. Such measures are typically questionable and arbitrary, which leads to some difficulty in comparing results. Nevertheless, any set of variables intended to be an authentic description of accessibility standards must be grounded in the literature and meet certain statistical criteria [56]. The latter refers, in particular, to measurability and reliability. In addition, the set should draw on spatially diverse and non-overlapping data. The report cited above was prepared for the purpose of central government reporting and, as such, represents the only source of relevant data—but one that fortunately happens to be very comprehensive. The only limitation related to potential difficulty in obtaining NUTS 2 data. As far as it is known, there has been no other research pertaining to this particular level. Therefore, the variables incorporated into the synthetic measure relate to various aspects of public entity accessibility, i.e., the architectural, the digital, and the technological (ICT). The final

number of variables was thus a function of usable variables provided in the report and their formal/statistical verification. This synthetic accessibility measure thus drew on the following components (Table 4).

**Table 4.** Set of variables included in the synthetic measure of accessibility.

| No. | Variable | Aspect |
|---|---|---|
| 1 | Percentage of public entities maintaining a website (by voivodeship) | (2) |
| 2 | Percentage of public entities providing a mobile app (by voivodeship) | (2) |
| 3 | Percentage of public entities providing an accessibility statement for all websites (by voivodeship) | (2) |
| 4 | Percentage of public entities providing an accessibility statement for all mobile apps provided (by voivodeship) | (2) |
| 5 | Percentage of websites and mobile apps fully compliant with the accessibility requirements set out in the Digital Accessibility Act (by voivodeship) | (3) |
| 6 | Percentage of public entities with barrier-free access on their premises (by voivodeship) | (1) |
| 7 | Percentage of public entities providing room layout information on their premises (by voivodeship) | (1) |
| 8 | Percentage of public entities allowing entry with guide dogs (by voivodeship) | (3) |
| 9 | Percentage of public entities providing a Polish Sign Language translator on their websites/mobile apps (by voivodeship) | (3) |
| 10 | Percentage of public entities providing a Polish Sign Language translator for face-to-face communication (by voivodeship) | (3) |
| 11 | Percentage of public entities providing a "what we do" notice in machine-readable text on their websites (by voivodeship) | (3) |
| 12 | Percentage of public entities providing a "what we do" notice in ETR (easy-to-read) text on their websites (by voivodeship) | (3) |

Source: Own elaboration. Aspects of accessibility: (1) the architectural; (2) the digital; (3) the technological.

## 3. Results and Discussion

The analysis shows that accessibility is highly variable among regions. The difference between upper- and lower-bound values of the synthetic accessibility measure exceeds 0.435 (Table 5). The highest value was noted for the Mazowieckie voivodeship (0.643)–three times higher than for the Podkarpackie voivodeship, which ranked the lowest among the Polish voivodeships (NUTS 2).

**Table 5.** Ranking of Polish voivodeships according to the accessibility of public sector bodies (as of 1 January 2021).

| No. | Voivodeship (NUTS 2) | Level of Synthetic Measure in 2021 |
|---|---|---|
| 1 | Mazowieckie | 0.643164 |
| 2 | Dolnośląskie | 0.577674 |
| 3 | Śląskie | 0.575463 |
| 4 | Łódzkie | 0.545997 |
| 5 | Kujawsko-Pomorskie | 0.543882 |
| 6 | Warmińsko-Mazurskie | 0.543677 |
| 7 | Opolskie | 0.522127 |
| 8 | Lubuskie | 0.511749 |
| 9 | Pomorskie | 0.493014 |

**Table 5.** *Cont.*

| No. | Voivodeship (NUTS 2) | Level of Synthetic Measure in 2021 |
|-----|----------------------|-------------------------------------|
| 10 | Wielkopolskie | 0.484241 |
| 11 | Podlaskie | 0.44378 |
| 12 | Zachodniopomorskie | 0.400517 |
| 13 | Lubelskie | 0.387565 |
| 14 | Małopolskie | 0.368698 |
| 15 | Świętokrzyskie | 0.308514 |
| 16 | Podkarpackie | 0.207801 |

Source: Own elaboration.

Some voivodeships were relatively close to each other in accessibility values, especially at the upper rungs of the ranking. The list is topped by the Mazowieckie voivodeship, which includes the Polish capital of Warsaw. Being a major metropolitan area and a large agglomeration, the capital often inflates the voivodeship's position in various indicators of socioeconomic development. The region is unique in that, apart from its leading urban hub, most of its areas are less socioeconomically developed. As such, the presence of the capital somewhat distorts the full picture of the region's status. Researchers of economic growth have noted that the region's situation would be very different if Warsaw were to be excluded [57]. Subsequent placements are populated by voivodeships with strong central hubs (Dolnośląskie; capital city—Wrocław) or high urbanization rates (Śląskie; capital city—Katowice). While the low placement of the Małopolskie voivodeship (capital city—Kraków) may be somewhat surprising, the relatively low urbanization rate in rural areas may contribute to a low-end ranking. On the other hand, the high placements of two eastern voivodeships are a welcome finding, as eastern-border regions tend to suffer from socioeconomic problems such as population ageing, depopulation, and relatively low population density. It turns out, however, that despite these challenges, accessibility of public entities in the Warmińsko-Mazurskie and Podlaskie voivodeships was found to be relatively solid. Unfortunately, the other eastern Polish provinces ranked among the worst in terms of their synthetic accessibility score. In particular, the Świętokrzyskie and Podkarpackie voivodeships clearly underperformed compared to the other regions. This is further supported by the four-group clustering presented in Table 6.

**Table 6.** Classification of public sector body accessibility in Polish voivodeships (NUTS 2) according to the synthetic accessibility measures.

| Class | Voivodeship (NUTS 2) |
|-------|----------------------|
| Class 1 | Mazowieckie, Dolnośląskie, Śląskie, Łódzkie, Kujawsko-Pomorskie, Warmińsko-Mazurskie |
| Class 2 | Opolskie, Lubuskie, Pomorskie, Wielkopolskie, Podlaskie |
| Class 3 | Zachodniopomorskie, Lubelskie, Małopolskie |
| Class 4 | Świętokrzyskie, Podkarpackie |

Source: Own elaboration.

The gap between these two voivodeships and the others is further illustrated by the last, smallest group generated by the distance-/extreme-based classification (8), a cluster consisting exclusively of these two regions. Furthermore, the top class comprised no less than six voivodeships, which had relatively high and homogenous values. In the differential (distance based) classification, the majority of the objects (11 out of 16) were grouped in the first two classes, indicating a wide gap between the top-ranking and lowest-ranking voivodeships. This problem was found to be the most pronounced for public sector bodies in south-eastern Poland. A visual illustration of this finding is presented in Figure 2.

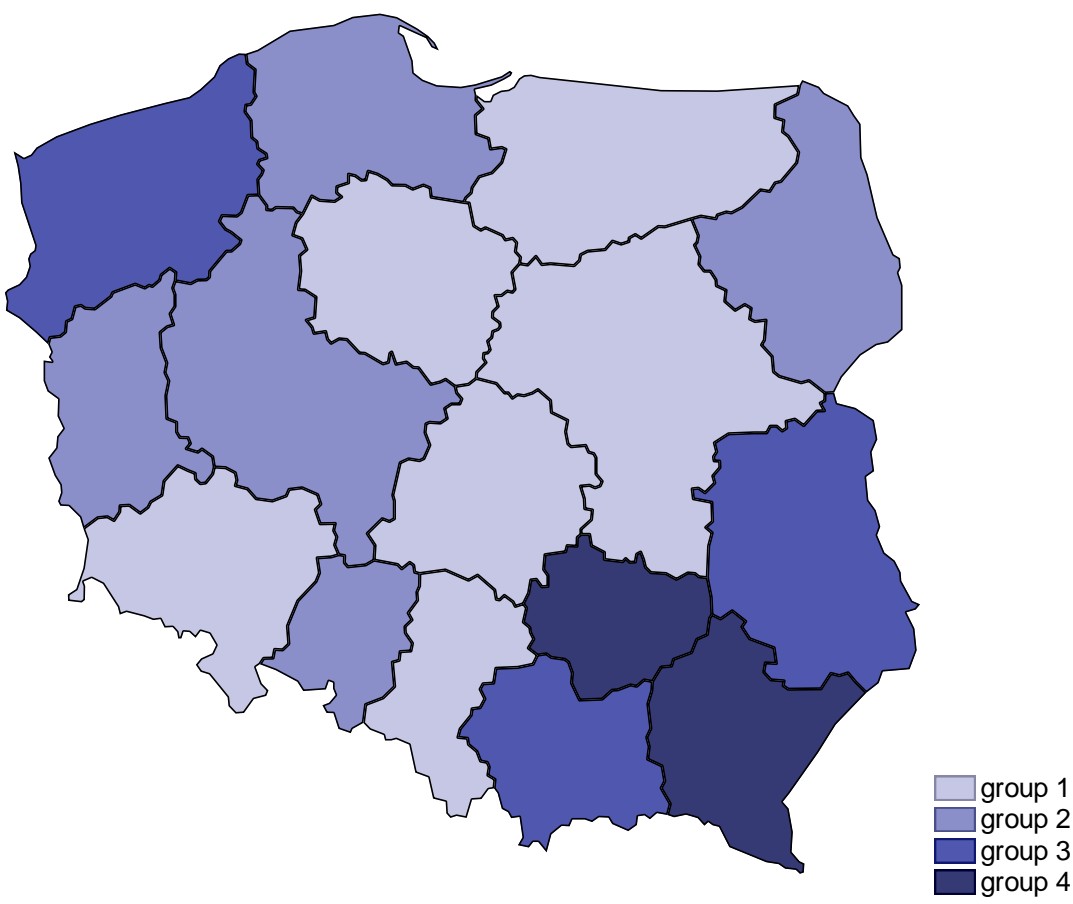

**Figure 2.** Public sector body accessibility in Polish voivodeships according to the synthetic accessibility measure (as of 1 January 2021).

As previously described, the last step of the analysis was to identify voivodeships (NUTS 2) that were the most homogenous in terms of public sector body accessibility. This step was designed to group voivodeships which are, relatively, the most alike. The aim was to identify those regions of Poland that require the most improvements in public body accessibility (divided, of course, into three types: physical, digital, and ICT accessibility). Ward's analysis points to the problematic situation of south-eastern regions. Seven clusters were identified (Figure 3), but their distribution was similar to that presented in Table 4. Based on the distribution of values and cluster averages (untested), it seems likely that the order of the voivodeships stems from decreasing values of diagnostic variables.

Notably, both the top-ranking voivodeship (Mazowieckie) and the bottom-ranking one (Podkarpackie) formed separate clusters of their own. There were thus the best- and worst-performing regions in terms of accessibility. The Wielkopolskie voivodeship (ranked 9th) served as a cut-off point of sorts, dividing the group into two (final) clusters.

As previously mentioned, the final placements in the accessibility ranking somewhat correspond to the socio-economic development of the voivodeships. The highest synthetic accessibility scores were attained by the most prosperous Polish voivodeships, including regions with major Polish urban centers (Warsaw, Wrocław, the Upper Silesian urban area, and Łódź). Others were not far behind, including the Tri-City area (Gdańsk, Sopot, and Gdynia), Poznań, and most of the western regions. These findings are in line with other research on digital exclusion ([15], p. 219), web accessibility ([16], p. 230), and well-being among inhabitants of these regions. Another crucial finding is the split of the eastern voivodeships into two groups despite their previously similar placements. The accessibility of public sector bodies, especially in the Warmińsko-Mazurskie voivodeship (capital city—Olsztyn) and, to a lesser extent, the Podlaskie voivodeship (capital city—Białystok) is a

welcome change from previous rankings of Polish regions, which placed them near the bottom with the other eastern regions.

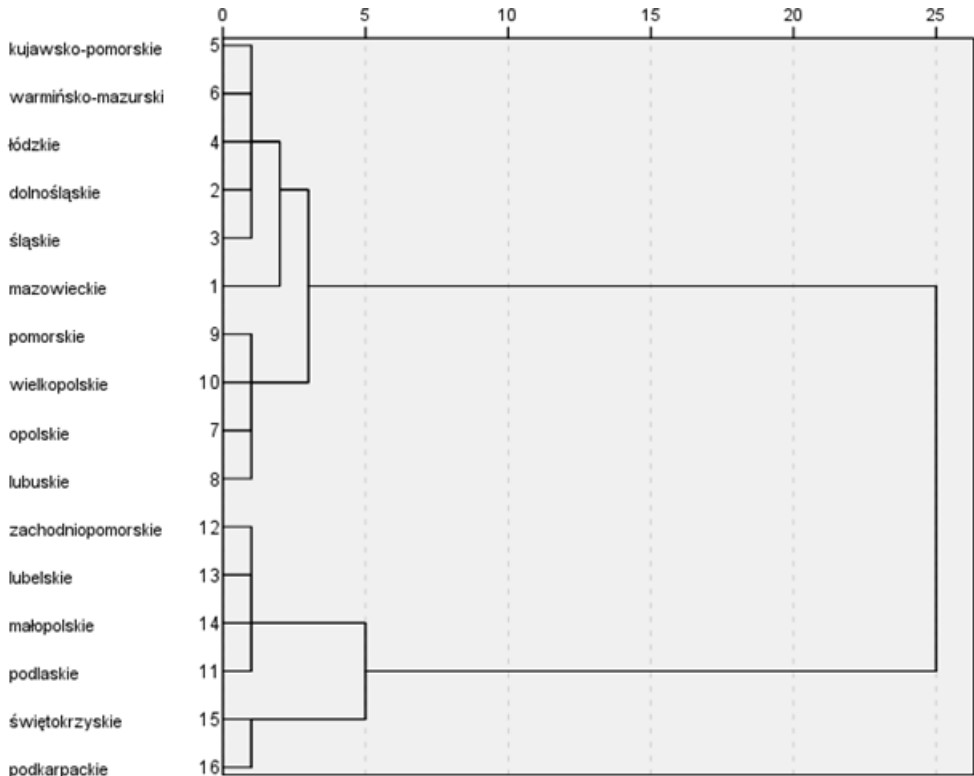

**Figure 3.** Dendrogram of similarity in public sector body accessibility (Ward's method, Euclidean distance, and scaled values) in Polish voivodeships (as of 1 January 2021).

The current method assigns a synthetic value (between 0 and 1) to each region. Assuming that a synthetic accessibility value close to 1 denotes complete accessibility of public sector bodies, then the actual results are sorely inadequate. In fact, even the best-performing localities have only reached approx. 65% of the optimal accessibility score. In the Podkarpackie voivodeship (capital city—Rzeszów), this percentage is no more than 20%, meaning that only one-fifth of the public sector bodies offer accessibility at all three levels (physical, digital, and ICT). This outcome certainly leaves much to be desired. After all, public sector bodies offer little in terms of web accessibility—a major hindrance, given the development of communication technologies, the ubiquity of the Internet, and the outbreak of the pandemic. Therefore, the digital accessibility of official documents and websites is important for the image and effectiveness of public entities. This is also the case for ICT accessibility. Though maintaining effective functioning (and thus accessibility) of public entities requires relatively few financial commitments, this subset of the diagnostic variables also produced unsatisfactory values. Such accessibility was usually introduced only when needed. Unfortunately, despite the accessibility requirements imposed by law, progress has been slow.

Providing physical accessibility entails considerable investment. Adaptation of existing buildings/premises and installation of new improvements involves expenditures that cannot always be secured in the short term. Nevertheless, this subset of the diagnostic variables also produced inadequate values.

Finally, we need to address the limitations of this study. First, despite the public sources of data and the comprehensive nature of the study, the diagnostic variables were screened to ensure availability for all voivodeships (NUTS 2). As such, the selection should be considered idiosyncratic and arbitrary, though also supported by a literature review and designed to be as comprehensive as possible. Secondly, it is reasonable to

expect that the current synthetic accessibility measure may produce other scores if other diagnostic variables are used. Therefore, these findings should be approached as just one contribution to the discussion on the design and components of synthetic accessibility measures. Again, however, it should be stressed that the report commissioned by The Ministry of Development Funds and Regional Policy, which takes into account the legal requirements imposed on public entities, is a reliable source of data.

## 4. Summary and Conclusions

Ensuring a relatively high level of accessibility of public sector institutions for people with special needs is undoubtedly one of the paths to sustainable socio-economic development. The current analysis of comprehensive and reliable data shows pronounced geographic differences in accessibility among Polish public sector bodies. A synthetic measure of public sector body accessibility was constructed for this study and used to rank Polish voivodeships (NUTS 2) from highest to lowest. Voivodeships with similar scores were identified using Ward's method. The Mazowieckie voivodeship (capital city—Warsaw) performed the best in terms of accessibility, though still below the requirements set by applicable legislation. The lowest accessibility score by far was noted for the Podkarpackie voivodeship (capital city—Rzeszów). These two extremes of the synthetic measure spectrum were outliers (the Podkarpackie voivodeship fared particularly poorly in this regard).

To conclude, actual accessibility among public entities was far from ideal in all respects (physical, digital, and ICT). Given the dangers of the pandemic, with many documents and issues having to be resolved electronically via technological channels, such accessibility should have become much more ubiquitous. Physical accessibility will probably be the most cost-intensive to provide, requiring strategic allocation of resources. Renovating buildings that currently offer little to no adaptations for the disabled will entail high expenditures or relocation. Improving digital and socio-communicative accessibility should be less problematic and, for the most part, limited to the universal implementation of WCAG.

This does not mean, however, that the indicated entities were not accessible at all. As noted, the most noticeable improvement was in architectural accessibility. Therefore, now more attention should be paid to improving digital and ICT accessibility. On this occasion, it is recommended that the increase in accessibility should be accompanied by improving the competences of the employees of these entities. At this level, soft skills seem crucial.

However, the best course of action for public entities would be to prepare a new accessibility report and show improvements. After all, this is imposed by official government documents—compliance with the requirements of such documents will be the best driver of accessibility improvement and expansion among Polish public sector bodies.

**Author Contributions:** Conceptualization, M.J., M.P. and E.K.; methodology, M.J., E.K. and K.M.; software, M.J.; validation, M.J., M.P., E.K and M.G.-B.; formal analysis, M.J.; investigation, M.J., M.P. and E.K.; resources, M.J, M.P. and E.K.; data curation, M.J., M.P., E.K. and M.G.-B.; writing—original draft preparation, M.J.; writing—review and editing, M.J, M.P., E.K. and M.G.-B.; visualization, M.J. and E.K.; supervision, M.J., M.P., E.K. and M.G.-B.; project administration, M.P.; funding acquisition, M.P. All authors have read and agreed to the published version of the manuscript.

**Funding:** Publication co-financed from the state budget under the program of the Minister of Education and Science (POLAND), called "Science for Society", project number NdS/536964/2021/2021. Amount of funding: PLN 1,557,100. Total value of the project: PLN 1,557,100.

**Institutional Review Board Statement:** Not applicable.

**Informed Consent Statement:** Not applicable.

**Data Availability Statement:** Publicly available datasets were analyzed in this study. This data can be found here: https://www.funduszeeuropejskie.gov.pl/media/106495/Raport_z_badania.pdf (accessed on 1 June 2023).

**Conflicts of Interest:** The funders had no role in the design of the study; in the collection, analyses, or interpretation of data; in the writing of the manuscript; or in the decision to publish the results.

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
