# Peer review of "Accessibility of Public Sector Institutions for People with Special Needs in Polish Regions"

_sustainability, doi:10.3390/su152215842_

Round 1

Reviewer 1 Report

Comments and Suggestions for Authors

The reviewed article is of a research nature. It's up to date. This is an interesting analysis of the public sector based on data from the reports. The research problem concerns the topic of accessibility of Polish public-sector bodies as a prerequisite for social inclusion. This topic is new and interesting. The authors used the available data. The authors made appropriate analyses. Three areas of accessibility were tested: physical (architectural) accessibility, digital (web) accessibility and ICT accessibility. The study I am reviewing is based on data from the “Status report on providing accessibility to people with special needs”. This document was prepared by the Ministry of Development Funds and Regional Policy and facilitated by the United Nations Convention on the Rights of Persons with Disabilities. I consider the whole work as an interesting cognitive study. The problem of public-sector body accessibility has not been researched so far to the extent presented by the authors of the article. The article does not require corrections. It contains an up-to-date bibliography. The structure of the text is correct. Summary and conclusions are also properly constructed.

Author Response

We would like to thank the Reviewer for useful comments.

Reviewer 2 Report

Comments and Suggestions for Authors

The topic of this study is extremely pertinent and up to date. The work has an appropriate structure, a suitable theoretical framework, however incipient, and a proper methodological approach. Congratulations to the authors. Below are some suggestions for improvement:

- In terms of the theoretical framework, it is suggested that the authors delve deeper into the three dimensions under study (physical accessibility, digital accessibility, and ICT accessibility), clearly highlighting the differences between the latter two. The focus of the explanation of these dimensions is very much on the last two, with little reference to the first (physical accessibility);

- In the introduction, the issue of accessibility in Poland should also be made clearer: what is the current state of play regarding the dimensions under study and why are the objectives of the study relevant?

- Graphically, in order to make it easier to understand, it is suggested that Table 1 be reformulated;

- In Table 2, to facilitate interpretation, we suggest grouping the 12 variables by the 3 dimensions under study (explaining which variables correspond to each dimension);

-In Figure 2, the authors should translate the map legend (grupa);

- In terms of discussion of the results, there are few links between the current study and studies already carried out on similar topics;

- The effective contributions of the study should be explained in point 4 (Summary and Conclusions).

Author Response

We would like to thank the Reviewer for every helpful and useful note.

  • - In terms of the theoretical framework, it is suggested that the authors delve deeper into the three dimensions under study (physical accessibility, digital accessibility, and ICT accessibility), clearly highlighting the differences between the latter two. The focus of the explanation of these dimensions is very much on the last two, with little reference to the first (physical accessibility);

  • Above-mentioned division into three dimensions is a result of the body of the report. And yes, we clearly highlighting the differences between the latter two because we are of the opinion that physical accessibility is well-described as all the obstacles and barriers with architectural and infrastructural issue.

  • - In the introduction, the issue of accessibility in Poland should also be made clearer: what is the current state of play regarding the dimensions under study and why are the objectives of the study relevant?

  • The issue of the accessibility of public-sector bodies isn't well-known issue both for scientists and policy makers. Therefore, it is very relevant and need more attention.

  • - Graphically, in order to make it easier to understand, it is suggested that Table 1 be reformulated;
    Thank you. Table 1 is now reformulated.

  • - In Table 2, to facilitate interpretation, we suggest grouping the 12 variables by the 3 dimensions under study (explaining which variables correspond to each dimension);
    There is an additional column in the Table 2 explaining which variables correspond to each dimension.

  • -In Figure 2, the authors should translate the map legend (grupa);
    Figure 2 is now fully translated.

  • - In terms of discussion of the results, there are few links between the current study and studies already carried out on similar topics;
    As mentioned before, there is a lack of study about accessibility in Poland. Therefore, any reports and article related to this topic were taken into account.

  • - The effective contributions of the study should be explained in point 4 (Summary and Conclusions). Relevant informations have been added.

Reviewer 3 Report

Comments and Suggestions for Authors

The study described in the present paper examined the accessibility of Polish public-sector 12 bodies based on data from government reports (comprehensive study). Accessibility is a feature that 13 should be offered as a complementary service offered to both individuals and legal entities during 14 epidemic emergencies and beyond.  (Can be combined into one objective statement. Objective statement should be concise and clear)

---One example is provided:

The objective of our study is to construct a synthetic accessibility measure that could show regional differences in accessibility for inclusion of people with special needs. 

--------------

Social inclusion has also 27 been cited as a major factor [1,2,3], ------for what ? 

 ICT ? information and communication technology (ICT)

voivodeships (provinces, NUTS 2 ??

--------------------

Legal accessibility requirements are the biggest driver of further 20 accessibility improvements for voivodeships.

------ do you want to say hurdles or drivers?

------------

Given that ICT solutions have become embedded in everyday life, it was necessary 60 to develop a theoretical framework to define and operationalize this problem.  (Is it problem or solution?) 

---------

These guidelines should be followed by public-sector instructions????

------------

“Status report on providing accessibility to 107 people with special needs” --- this changes the objective statement completely

For the purposes of the present paper, accessibility is defined as a feature of an envi- 131 ronment which enables persons with functional and cognitive difficulties to benefit from 132 the environment on an equitable basis, which is crucial for social and economic engage- 133 ment. P

-------------------

 Re-do Table 1 - Information can be presented in a better way

----------------

The synthetic accessibility measure was calculated using TOPSIS [49], commonly 145 used in economics research [50, 51, 52]. The method weighs decision alternatives by meas- 146 uring their distance from two reference points-------justify why the method is used ?

------------------------

To our best 217 knowledge, there has been no other research pertaining to this particular level. ???

---------------

The analysis shows that accessibility is highly variable between regions.

-among regions

-----------

 relatively, the most alike. The 275

served as a cut-off point of sorts, dividing the group into two (final) clusters

Comments on the Quality of English Language

Please see the notes to the author above.

At many places, the language can be concise. At couple of places authors are assuming things and using abbreviations. There seems to be different writing styles in the paper. At many places the writing has errors and at many places, it is excellent.

Reviewer 4 Report

Comments and Suggestions for Authors

The authors of the manuscript made an attempt to interpret the results of the “Report on the state of ensuring accessibility for people with special needs by public institutions in Poland” available at the website of the Polish Ministry of Development Funds and Regional Policy https://www.funduszeeuropejskie.gov.pl/media/106495/Raport_z_badania.pdf

Link given by authors (https://www.gov.pl/web/finanse/raport-o-stanie-zapewniania-dostepnosci-podmiotu-publicznego) doesn’t work.

Authors declare that the publication was co-financed from the state budget under the program of the Minister of Education and Science called "Science for Society" project number NdS/536964/2021/2021 “Companion robot navigation as a tool to improve the quality of life of people with limited mobility”. The content of this manuscript does not apply to this project.

The topic of the manuscript is much wider than its content. At least the topic should be formulated more precisely, for example, “Accessibility of public-sector institutions for people with special needs in Polish regions” or similar.

The main idea of the manuscript was to elaborate a “synthetic measure” of accessibility. The authors used the Technique for Order of Preference by Similarity to Ideal Solution (TOPSIS) to calculate the synthetic accessibility measure for Polish regions. TOPSIS is a multi-criteria decision analysis method, which is based on the concept that the chosen alternative should have the shortest geometric distance from the positive ideal solution and the longest geometric distance from the negative ideal solution. There is no explanation in the manuscript why this decision analysis method was used as a “synthetic measure” of accessibility.

The main problem of the application of this method appears at stage 3 (calculation the weighted normalized decision matrix). In accordance with the above-mentioned report, authors used 12 “variables included in the synthetic measure of accessibility” mentioned in Table 2. Every criterion (variable) should be weighted. Values of these weights should be given in the manuscript, and the process of calculation of these weights should be explained. The weights of the criteria in the TOPSIS method can be calculated using Ordinal Priority Approach, Analytic hierarchy process, etc.

It would be the important added value of this manuscript to propose and verify a method of quantitative comparing of different types of accessibility: physical, digital, and ICT. In the above-mentioned report, all types of accessibility are calculated and presented separately. The content is clear. In the manuscript calculation of synthetic measure’s values for Polish regions is not clear (Table 3).

It is recommended to reduce Figure 1 presenting copied from the original report “characteristics of the survey” and correct in line 126 the number of Polish institutions (the proper number is 57574).

Comments on the Quality of English Language

The text seems to be machine-translated from Polish. 

Round 2

Reviewer 3 Report

Comments and Suggestions for Authors

I still do not like the abstract. Your contribution needs to be highlighted. You have written as to what you did but not why. Objective statement is not strong enough. 

Tables 1 and 2 still need to be improved. Need to find a better way for presentation on Tables.

"The effective contributions of this study allow institutions to develop actions to 356 change this situation. However, the best course of action for public entities would be to 357 prepare a new accessibility report and show improvements. After all, this is imposed by 358 official government documents – compliance with the requirements of such documents 359 will be the best driver of accessibility improvement and expansion among Polish public- 360 sector bodies."

Conclusions are too broad and not directly related to study. If redoing everything is the suggestion, it could be done without the need of the study. Please try to link the contribution of the study to study if feasible. 

Comments on the Quality of English Language

As mentioned above "Abstract" still needs to be improved. 

Still do not like the Tables 1 and 2 presentation

Reviewer 4 Report

Comments and Suggestions for Authors

Comparing the actual version of the manuscript with its previous version, the authors made minor changes, mainly in language.

Despite the declaration of changing the title of the manuscript, the actual title of the manuscript remained unchanged. The topic of the manuscript still is much wider than its content. At least the topic should be formulated more precisely, for example, “Accessibility of public-sector institutions for people with special needs in Polish regions” or similar.

The authors of the manuscript still tried to interpret the results of the “Report on the state of ensuring accessibility for people with special needs by public institutions in Poland” available at the website of the Polish Ministry of Development Funds and Regional Policy

The content of the manuscript still has no substantive connection with the project NdS/536964/2021/2021 “Companion robot navigation as a tool to improve the quality of life of people with limited mobility” co-financed from the state budget under the program of the Minister of Education and Science called "Science for Society". Costs of publication of the manuscript cannot be financed from the budget of this project.

The authors still did not explain why they used the Technique for Order of Preference by Similarity to Ideal Solution (TOPSIS) to calculate the synthetic accessibility measure for Polish regions. TOPSIS is a multi-criteria decision analysis method, which is based on the concept that the chosen alternative should have the shortest geometric distance from the positive ideal solution and the longest geometric distance from the negative ideal solution. The main idea of the manuscript was to elaborate a “synthetic measure” of accessibility. The authors did not explain, what they understand in their calculations (not shown in the manuscript) as “the positive ideal solution” and “the negative ideal solution”. These terms have nothing to do with accessibility.

Justified concerns are raised not only by the choice of the research method, but also by the correctness of its application. In lines 151-151 of the manuscript, the authors wrote: “The procedure is performed in several steps. The first step is to build a decision matrix X=[xij] and calculate the weight vector w = [w1, …, wn], where w1 + …+ wn = 1”. As I wrote in my review, the main problem of the application of TOPSIS method appears at the stage of calculation of the weighted normalized decision matrix. In accordance with the above-mentioned report, authors used 12 “variables included in the synthetic measure of accessibility” mentioned in Table 2. Every criterion (variable) should be weighted. Values of these weights should be given in the manuscript, and the process of calculation of these weights should be explained. The weights of the criteria in the TOPSIS method can be calculated using Ordinal Priority Approach, Analytic hierarchy process, etc. In response to the review, the authors write “It was not decided to assign weights to individual variables”. So, how they calculated the weight vector w = [w1, …, wn], where w1 + …+ wn = 1?

Summing up, more reliable information about the accessibility of public-sector institutions for people with special needs in Polish regions is contained in the original report rather than in its interpretation by the authors of the manuscript. In the above-mentioned report, all types of accessibility are calculated and presented separately. The content of the report is clear.

Round 3

Reviewer 4 Report

Comments and Suggestions for Authors

Comparing the third version of the manuscript with its previous version, the authors made minor changes. 

The main idea of the manuscript is to add new value to the results of the “Report on the state of ensuring accessibility for people with special needs by public institutions in Poland” available at the website of the Polish Ministry of Development Funds and Regional Policy: https://www.funduszeeuropejskie.gov.pl/media/106495/Raport_z_badania.pdf The authors try to calculate the synthetic accessibility measure of public sector institutions for people with special needs for Polish regions. Their attempt failed. More reliable information about the accessibility of public sector institutions for people with special needs in Polish regions is contained in the original report rather than in its interpretation by the authors of the manuscript. In the above-mentioned report, all types of accessibility are calculated and presented separately. The content of the report is clear. 

The authors cannot explain why they used the Technique for Order of Preference by Similarity to Ideal Solution (TOPSIS) to calculate the synthetic accessibility measure for Polish regions. TOPSIS is a multicriteria decision analysis method, not a method to calculate accessibility. TOPSIS is based on the concept that the chosen alternative should have the shortest geometric distance from the positive ideal solution and the longest geometric distance from the negative ideal solution. The authors did not explain, what they understand in their calculations (not shown in the manuscript) as “the positive ideal solution” and “the negative ideal solution”. These terms have nothing to do with accessibility.

The authors seem to falsify or made serious mistake in their calculations. In their response to the previous review they wrote: "Writing that it was not decided to assign weights meant that each variable received a weight of 1". This sentence was intended to contain a response to the objection in the previous review, how the authors build a decision matrix X=[xij] and calculated the weight vector w = [w1, …, wn], where w1 + …+ wn = 1. In accordance with the above-mentioned report, the authors used 12 “variables included in the synthetic measure of accessibility”. So, if "each variable received a weight of 1", the weight vector w = 12. This is a serious mistake. In this situation, the results of further calculations are completely unreliable. 

The content of the manuscript has no connection with the "companion robot for people with limited mobility". The cost of publication of the manuscript cannot be financed from the budget of the project NdS/536964/2021/2021 “Companion robot navigation as a tool to improve the quality of life of people with limited mobility” co-financed from the state budget under the program of the Polish Minister of Education and Science.
